# Exploiting the role of nanoparticle shape in enhancing hydrogel adhesive and mechanical properties

Maria C. Arno [1✉], Maria Inam[2], Andrew C. Weems[1], Zehua Li[1,2], Abbie L.A. Binch[3], Christopher I. Platt [3], Stephen M. Richardson [3], Judith A. Hoyland[3,4], Andrew P. Dove[1✉] & Rachel K. O'Reilly [1✉]

The ability to control nanostructure shape and dimensions presents opportunities to design materials in which their macroscopic properties are dependent upon the nature of the nanoparticle. Although particle morphology has been recognized as a crucial parameter, the exploitation of the potential shape-dependent properties has, to date, been limited. Herein, we demonstrate that nanoparticle shape is a critical consideration in the determination of nanocomposite hydrogel properties. Using translationally relevant calcium-alginate hydro-gels, we show that the use of poly(*L*-lactide)-based nanoparticles with platelet morphology as an adhesive results in a significant enhancement of adhesion over nanoparticle glues comprised of spherical or cylindrical micelles. Furthermore, gel nanocomposites containing platelets showed an enhanced resistance to breaking under strain compared to their spherical and cylindrical counterparts. This study opens the doors to a change in direction in the field of gel nanocomposites, where nanoparticle shape plays an important role in tuning mechanical properties.

[1] School of Chemistry, University of Birmingham, B15 2TT Edgbaston, Birmingham, UK. [2] Department of Chemistry, University of Warwick, Gibbet Hill Road, CV4 7AL Coventry, UK. [3] Division of Cell Matrix Biology and Regenerative Medicine, School of Biological Sciences, Faculty of Biology, Medicine and Health, The University of Manchester, Manchester Academic Health Science Centre, Stopford Building, Oxford Road, Manchester M13 9PT, UK. [4] NIHR Manchester Biomedical Research Centre, Central Manchester Foundation Trust, Manchester Academic Health Science Centre, 29 Grafton Street, Manchester M13 9WL, UK. ✉email: m.c.arno@bham.ac.uk; a.dove@bham.ac.uk; r.oreilly@bham.ac.uk

Living systems are characterized by a diverse range of shapes[1]. From spherical HIV to rod-shaped tobacco mosaic viruses, star-shaped bacteria, and multi-shaped cells that constitute our body, morphology is known to grant evolutionary advantages based on specific functions, such as interaction with surfaces, passive diffusion, and active motility[2]. In an attempt to mimic Nature's perfection, scientists have designed a variety of nanoparticles, from carbon-based[3] or polymeric[4], to inorganic[5] or metal-based materials[6]. The possibility to precisely control the physicochemical properties of polymeric nanoparticles, such as shape, size, and surface chemistry has offered the opportunity to tailor these nanostructures to the desired application[7]. For example, nanoparticle morphology is one of the main factors that modulate the rate and mechanism of cellular uptake, as well as the intracellular transport[8–10]. Furthermore, shape has also been used to modulate the antimicrobial effect of silver nanoparticles[11].

Inspired by the significant influence nanoparticle shape has on drug delivery[12], cell–nanoparticle interaction and internalization rate[13,14], and even cell behavior or differentiation of stem cells[15], we hypothesized that the mechanical properties and adhesion between hydrogel materials could be controlled by tuning nanoparticle shape. A range of nanoparticles have previously been successfully combined with polymeric networks to obtain nanocomposite (NC) hydrogels, where the nanostructures physically or covalently interact with the polymeric chains and lead to enhanced mechanical properties compared to conventional natural polymer hydrogels, thus improving their interaction and performance in a biological environment[16–20]. Beyond enhancing preexistent hydrogel properties, additional features can also be exploited using gel NCs, including adhesion to surfaces[21]. In this direction, silica[22,23] or polymer-based[24] nanoparticles have been employed as adhesives between hydrogels and biological tissues. To date, only spherical nanoparticles have been studied for these applications; however, considering morphology strongly influences nanoparticle behavior, we postulated that a difference in shape would result in a different interaction with the gel matrix.

Herein, we seek to exploit the influence of nanoparticle shape on material properties by preparing polymeric nanoparticles using crystallization-driven self-assembly (CDSA), a methodology that allows high levels of control over particle size, morphology, and surface chemistry. In turn, using translationally relevant calcium-alginate hydrogels, we demonstrate that 2D platelets significantly increase both the adhesion between hydrogel surfaces and the material's mechanical strength, when blended into the polymeric network, compared to their 0D spherical or 1D cylindrical counterparts.

## Results

### Preparation of PLLA$_{35}$-based nanoparticles.
Neutral, cationic, anionic, and zwitterionic 2D platelets and cationic 1D cylinders were prepared from poly(L-lactide)-based block copolymers using CDSA methods[25] as previously reported[26] (Supplementary Tables 1–3, Fig. 1b, c, Supplementary Figs. 1–10). Spherical micelles were prepared by a solvent switch method in order to prevent core block crystallization that would lead to deviation from spherical shape, while retaining the same chemistry as for other particles (Fig. 1a, Supplementary Fig. 11, Supplementary Tables 2 and 3). This precise control over the shape and dimension of the nanoparticles, together with the possibility to obtain different morphologies using the same chemistry, open the doors to the investigation of NC properties based only on particle shape. Hypothesizing that different nanoparticle morphologies have different affinity and ability to interfere with network chains, we sought to investigate whether the adhesive energy between two

freshly cut parts of the alginate gel would be dependent on the shape of the nanoparticle applied as an adhesive.

### Adhesive properties of calcium-alginate hydrogels.
Calcium-alginate hydrogels prepared by mixing together alginate (1.5 wt%), calcium carbonate, and D-glucono-δ-lactone (GDL) were chosen as the preferred matrix to explore NC properties as a consequence of their ease of preparation and translational relevance in the biomaterials field. A dispersion of the quaternized (cationic) platelets in water at 30 mg mL$^{-1}$ was used as an adhesive to glue together two calcium-alginate hydrogel blocks. A 10 µL drop of the solution was spread on the hydrogel surface (ca. 5 mm × 10 mm), as glue, and another hydrogel block was placed on top (Fig. 2a–d, Supplementary Table 4). Adhesion was observed within 10 min by lifting the top block. Additional blocks were glued in the same way to create a bridge that was strong enough to hold when suspended horizontally between two posts or vertically from a needle (Fig. 2b, Supplementary Fig. 12). Negligible adhesion was observed when calcium-alginate gels were used on their own, or when water or a quaternized homopolymer solution was used as a glue substitute, demonstrating that the presence of quaternized platelets at the interface is responsible for the observed adhesive property. Additionally, the adhesion remains when the joint is immersed in an excess of water and swells (Supplementary Fig. 13). To further explore this, the effect of size and surface charge on the adhesive properties of the platelets was evaluated (Fig. 2f, g). Firstly, small (372 nm (length) × 223 nm (width)), medium (893 × 528 nm), and large (1700 × 993 nm) platelets were compared, with small platelets showing a higher adhesion strength compared to medium and large platelets (Fig. 2f, Supplementary Table 5). We postulate that this size-dependent behavior is a consequence of the increased ability of the small platelets to both lie on the contours of the gel surface and not interact with each other to form a better adhesive layer than larger platelets, which in turn tend to overlay on top of each other, thus limiting their interaction with the gel surface. Secondly, the effect of charge was evaluated (Fig. 2g, Supplementary Table 6). Quaternized platelets provided a higher adhesion energy compared to anionic, zwitterionic, and neutral platelets, likely as a consequence of their better interaction with the anionic un-cross-linked alginate groups.

The adhesion energy of these platelet particles was then compared with the 0D spherical and 1D cylindrical morphologies. As expected, it was found that spherical and cylindrical micelles also exhibited adhesion behavior under these conditions. However, quantitative measurements revealed that the platelet-glued gels could withstand 133% and 600% more stress before breakage when compared to the corresponding spherical and cylindrical particle-glued gels (1400 vs 600 and 200 Pa, respectively) at the same overall polymer loading (0.12 wt%; Fig. 2c). Furthermore, 2D platelet nanoparticles display an enhancement in uniaxial tensile adhesive energy ca. nine times higher that of the spherical particles (1.1 J m$^{-2}$) and ca. four times over cylindrical micelles (2.2 J m$^{-2}$), as assessed by measuring the adhesive energy of bulk shear and uniaxial tensile testing of two freshly cut calcium-alginate gel blocks (Fig. 2c, d, Supplementary Table 4). Statistical analysis of the bulk shear adhesive energy confirmed the significantly superior adhesion provided by platelets compared to spherical and cylindrical micelles (Supplementary Table 7). Importantly, in order to demonstrate that the total particle surface area does not influence the overall adhesive energy, three times more spheres were applied to increase the total surface area of the particles such that it was comparable with that of the platelets. This resulted in a substantially reduced adhesive energy (Supplementary Fig. 14), which further validates

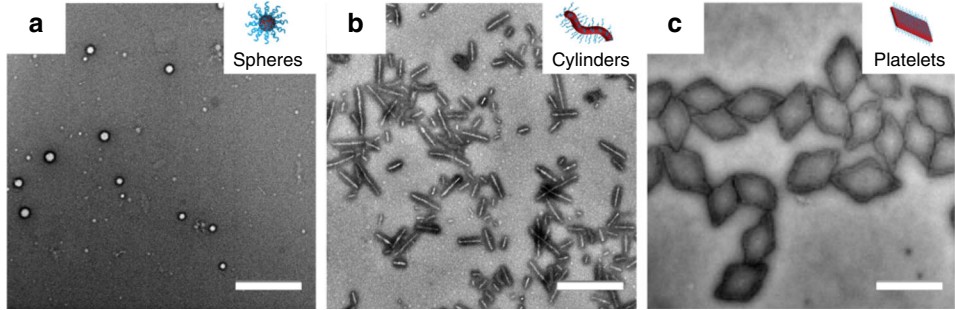

**Fig. 1 Characterization of PLLA$_{35}$-based nanoparticles of different morphologies.** Representative TEM micrographs of **a** PLLA$_{35}$-*b*-PDMAEMA$_{315}$ spherical micelles prepared via a solvent switch from *N,N*-dimethylformamide to water, ($D_h = 149$ nm, PD = 0.245 as obtained by dynamic light scattering), **b** PLLA$_{35}$-*b*-PDMAEMA$_{315}$ platelets (length = 893 ± 100 nm, width = 528 ± 60 nm as measured using ImageJ from TEM micrographs) prepared via CDSA, heating the polymer solution in EtOH at 90 °C for 2 h, and **c** PLLA$_{35}$-*b*-PDMAEAm$_{400}$ cylindrical micelles prepared via CDSA (length = 300 ± 52 nm, width = 10 ± 1 nm as measured using ImageJ from TEM micrographs). Samples were stained with 1% uranyl acetate. Scale bar = 1000 nm for **a** and **c**, 500 nm for **b**.

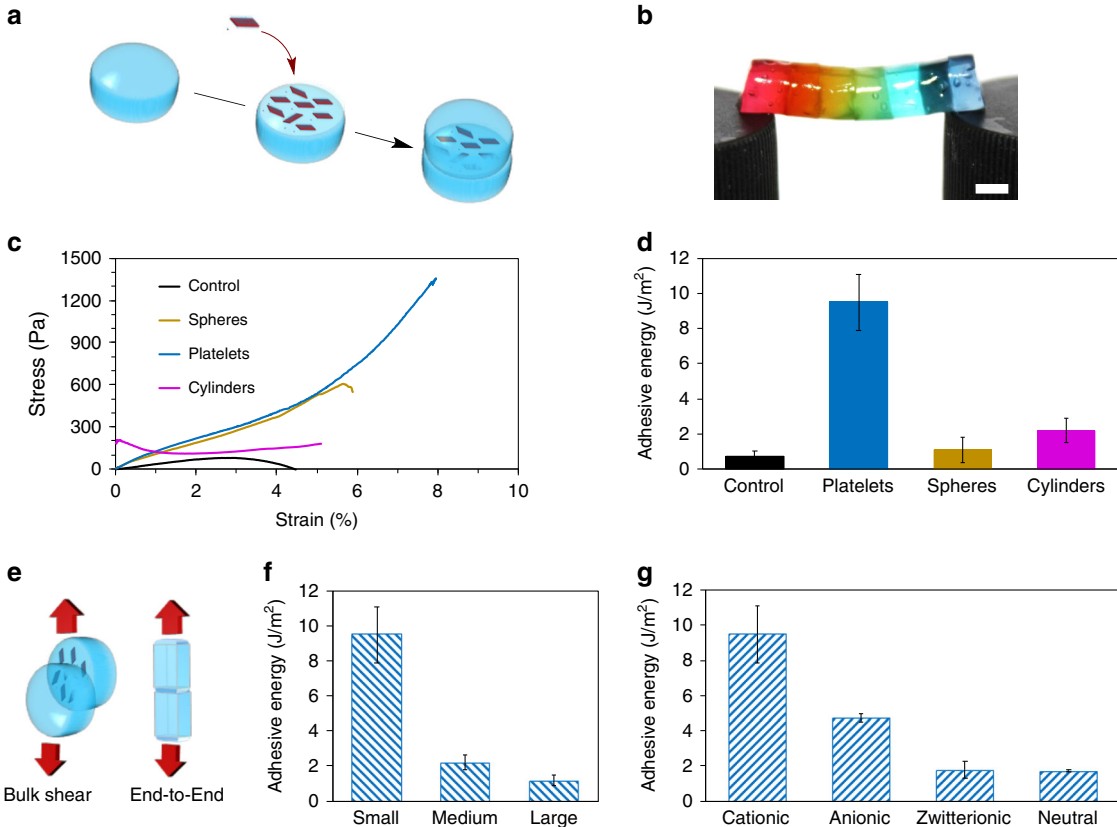

**Fig. 2 Adhesion properties of calcium-alginate hydrogel blocks glued using PLLA$_{35}$-based nanoparticles of different morphologies. a** Schematic representation of alginate hydrogel blocks adhesion using quaternized platelets. **b** Photograph of calcium-alginate hydrogel blocks glued together with PLLA$_{35}$-*b*-PDMAEMA$_{315}$ platelets suspended horizontally. Scale bar = 0.5 cm. **c** Bulk shear stress behavior comparing representative interfacial stresses and **d** Adhesive energy of bulk shear of two calcium-alginate identical gel blocks adhered with water (control), platelets, cylindrical, or spherical micelles. Values have been normalized for total surface area. **e** Schematic representation of end-to-end and bulk shear adhesion experiments performed with DMA to quantify adhesion. Effect of size **f** and charge **g** on the adhesive energy of PLLA$_{35}$-*b*-PDMAEMA$_{315}$ platelets (small: 372 × 223 nm, medium: 893 × 528 nm, and large: 1700 × 993 nm). Error bars represent the standard deviation of the data. Statistical analysis of these data can be found in Supplementary Tables 5–7.

the important contribution of particle shape (and not simply surface area) to the adhesion energy, and provides a method by which to create a robust adhesive at low polymer loading by simply manipulating the nanoparticle morphology.

**Mechanical properties of NC alginate hydrogels.** Encouraged by the promising role of nanoparticle shape in determining the

adhesion strength between calcium-alginate hydrogels blocks, we sought to investigate whether nanoparticle morphology could also be used to tune the mechanical properties of translationally relevant alginate NCs. The hydrogels were prepared as described earlier and readily modified by incorporating various amounts of platelet, cylindrical, or spherical particles at room temperature with a constant concentration of calcium to aid cross-linking.

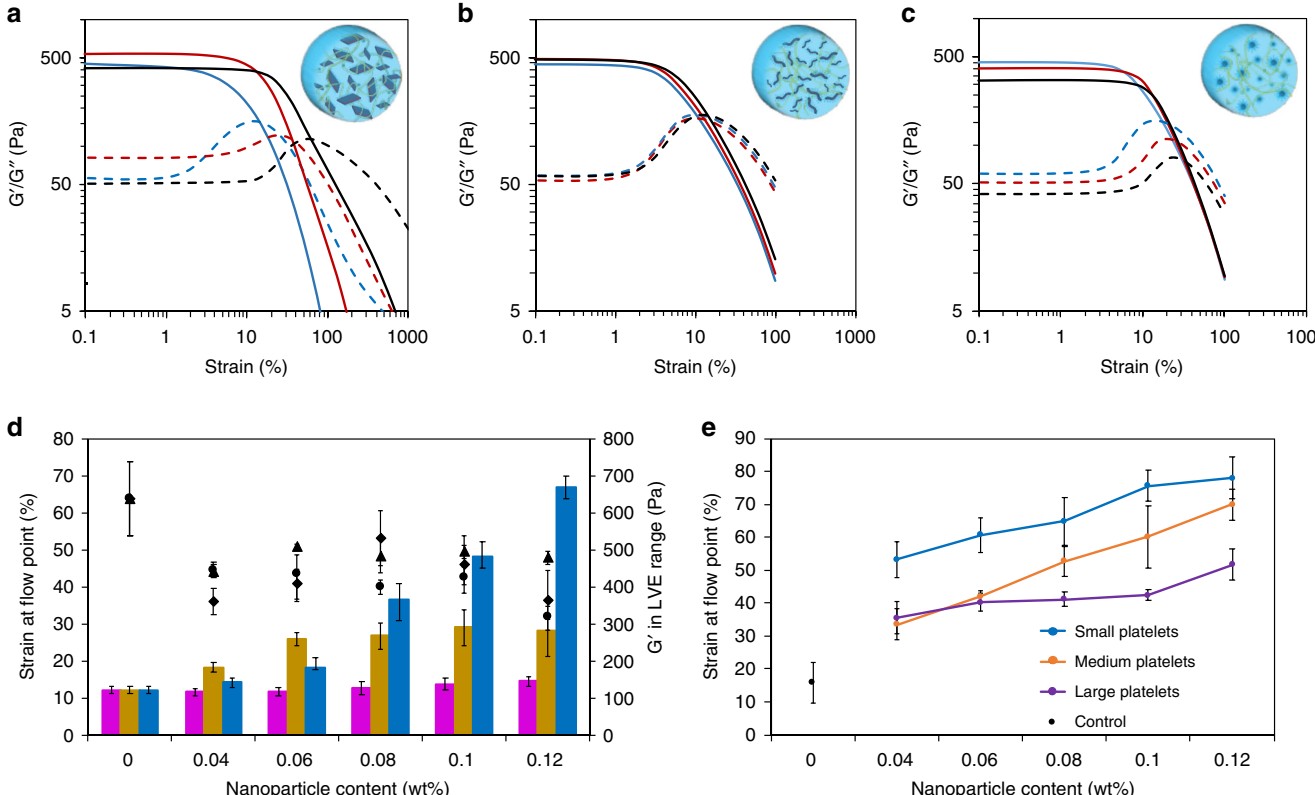

**Fig. 3 Rheological properties of calcium-alginate hydrogels enriched with PLLA₃₅-based nanoparticles of different morphologies.** Strain-dependent oscillatory rheology measurements showing storage modulus (G′, solid line) and loss modulus (G″ dashed line) of NC calcium-alginate hydrogels with 0.04 (blue), 0.08 (red), and 0.12 wt% (black) platelets (1124 × 676 nm) **a**, cylindrical **b**, and spherical micelles **c**. **d** Comparison of strain at the flow point (Y-axis, left) and G′ in the linear viscoelastic region (Y-axis, right) for NC hydrogels incorporating platelets (blue bar for G′, ♦ for strain at flow point), spherical (dark yellow bar, ●), or cylindrical micelles (magenta bar, ▶) at different wt% (X-axis). **e** Comparison of strain at the flow point between different sizes of platelets (small: 372 × 223 nm, medium: 893 × 528 nm, and large: 1700 × 993 nm) as a function of the total surface area. Error bars represent the standard deviation of the data. Statistical analysis of these data can be found in Supplementary Tables 9 and 10.

Oscillatory rheology measurements were performed, in order to quantify gel strength. A broad linear viscoelastic region and network failure at high strains was observed for all gels in multiple runs with the frequency dependence of the storage and loss moduli (G′ and G″, respectively), confirming gel-like behavior as G′ > G″ in the entire range of frequencies observed (Supplementary Fig. 15). Upon gelation in the presence of spherical micelles, strain-dependent oscillatory rheology showed a small increase in strain at the flow point (crossover point of G′ and G″) such that the materials flowed at ~25% strain at break, compared to ~10% when no additive was used (Fig. 3c, d, Supplementary Fig. 16); however the change in G′ at low strain was minimal. A strain at break of 15% was measured by oscillatory rheology for hydrogels containing cylindrical micelles (Fig. 3b, d), with a minimal change compared to the gels with no nanoparticle additive. This is not surprising, as the elongated morphology of cylindrical micelles is expected to interact less with the alginate gels, as a consequence of the lower surface area per particle ($9.6 \times 10^{-11}$ vs $7.1 \times 10^{-10}$ cm$^{-2}$, Eqs. (1)–(11)), leading to a lower strain at break.

Investigation of the strain-dependent oscillatory rheology of the platelet-containing gels showed a substantial increase in strain at the flow point in comparison to those containing spherical or cylindrical constructs (Fig. 3a–d), which demonstrates that the 2D platelet-containing hydrogel was able to withstand higher shear strain before failure (up to ~70% strain at break) than the spherical and cylindrical-micelle-containing hydrogel at comparable micelle loading (Fig. 3a–d). Statistical analysis of the strain at

flow point at different nanoparticle loading confirmed that the enhancement of the mechanical properties provided by platelets is significantly higher compared to spherical and cylindrical micelles (Supplementary Table 8). Furthermore, the strain at yield (initial drop in G′) also increased upon increasing the platelet content (Fig. 3a), which was not observed for the addition of spherical or cylindrical micelles (Fig. 3d). This can be ascribed to the higher interaction of the 2D platelets with the alginate hydrogel, hence the more platelets are incorporated in the gel, the higher the strain at flow point. The minimal effect of both spherical and cylindrical micelles leads to no significant increase in strain at flow point, even at a higher amount of nanoparticles in the hydrogel system. As expected, a maximum limit was reached (0.12 wt%) where the gels showed decreased homogeneity, presumably as a consequence of the increased competition between micelle-alginate and calcium-alginate ionic interactions becoming important and hence disrupting gel formation. These observations confirm the importance of nanoparticle additive shape on the mechanical resistance to shear. We postulate that, as a consequence of their 2D morphology, platelets offer a greater surface area that is more easily available to interact with the hydrogel's polymeric chains, therefore further enhancing the mechanical properties of alginate gels compared to both spherical and cylindrical morphologies. Importantly, no substantial change in G′ in the linear viscoelastic range was noted with increasing content of platelets, spherical or cylindrical structures, thus avoiding potential embrittlement of the materials with the incorporation of these additives. Moreover,

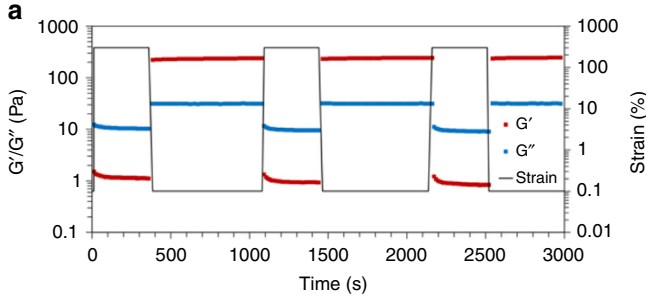

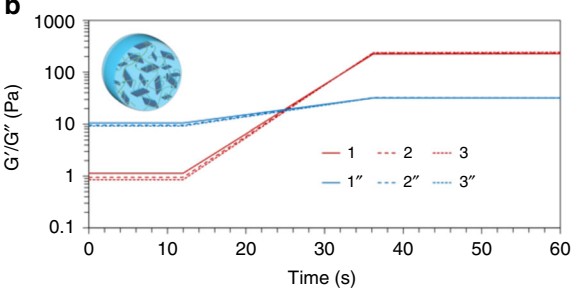

**Fig. 4 Self-healing properties of calcium-alginate hydrogels containing PLLA$_{35}$-*b*-PDMAEMA$_{315}$ platelets. a** Step strain measurements of NC calcium-alginate containing platelets (0.12 wt%) over three cycles ($\omega = 10$ rad s$^{-1}$) and **b** overlaid zoom of the recovery of the material properties after each of three cycles, indicated by G' in a continued line and G" in a dashed line (the numbers indicate the repeats).

as previously observed for the adhesion energy, small (372 × 223 nm) platelets enhance the material properties more compared to large (1700 × 993 nm) or medium (893 × 528 nm) platelets, even at similar total surface area, likely as a consequence of a better dispersion of small platelets in the hydrogel structure (Fig. 3e, Supplementary Tables 9 and 10).

To further demonstrate that the change in nanoparticle shape can be ascribed to only influence the mechanical (yield and breaking) strength and not the stiffness of the hydrogel materials, we studied the reversible shear-thinning and self-healing behavior of the non-covalently cross-linked hydrogels, which are important for the potential injection of the materials to their site of ultimate use. Despite the increase in strength upon platelet addition, fully reversible induction of flow under applied shear stress, followed by rapid self-healing upon relaxation of the stress was retained (Fig. 4a, b). The retention of these properties alongside increased resistance to yield and breaking overcomes one of the major impediments to wider study and application of calcium-alginate hydrogels in tissue engineering approaches, while retaining their most attractive features.

**Cytocompatibility and tissue adhesion of NCs.** Seeking to use the reinforced and adhesive alginate hydrogels for tissue engineering applications, the cytocompatibility of our biomaterials was assessed by 3D encapsulation of adipose-derived stem cells (ASCs) over a period of 7 days. Cell viability in calcium-alginate hydrogels containing platelet particles was similar to particle-free calcium-alginate hydrogels after 3 and 7 days; no statistical differences were observed, which demonstrates that the materials are highly cytocompatible (Fig. 5c). Furthermore, the ability of our platelets to adhere cell-laden hydrogels was investigated under physiological conditions. A construct with alternate layers of encapsulated ASCs gel and acellular gel was built using the PLLA$_{35}$-*b*-PDMAEMA$_{315}$ platelets as an adhesive between layers. As expected, the adhesion site remained intact after hydrogel swelling when immersed in cell culture medium (Supplementary

Fig. 17) at 37 °C. Interestingly, no cell migration was observed from the cell layer to the acellular layer (Fig. 5a), which suggests that our system has the potential to be used to seal together different tissues, an essential property to ensure wound healing and tissue regeneration. To further investigate this possibility, we then encapsulated dye-labeled chondrocytes and dye-labeled osteoblasts in two different calcium-alginate hydrogel blocks, which were then glued together using the platelet particles to provide a simplistic mimic of the osteochondral junction. In accordance with the results reported above, no cell migration between the two layers was observed after 7 days (Fig. 5b). These advances suggest that the use of platelet particles as adhesives provides a promising route to enable the construction of complex tissue engineering models, using calcium-alginate hydrogels for regeneration of different tissues. Finally, the ability of the platelets to be used as a glue was also explored using bovine cartilage tissue. The tissue was cut and immediately covered with the platelet solution and left adhered for 2 h. The adhesive energy was then measured with a lap shear test, showing a shear strength of 66 ± 18 kPa, which is above the minimum threshold for clinical applications in repair of meniscal tears[27], although lower than native cartilage tissue (433 ± 360 kPa; Fig. 5). Nevertheless, the adhesive energy provided by our platelets is higher than the adhesive energy reported for PEG-DOPA and commercial fibrin adhesive Tisseel (35 ± 12.5 and 2.58 ± 1.76 kPa, respectively), although lower than the commercial cyanoacrylate Dermabond (181 ± 33.4 kPa)[28,29].

## Discussion

Clearly, the shape of nanoparticles plays a key role in the definition of the resultant properties of hydrogel materials, as well as in the strength of adhesion when used as a glue. Indeed, harnessing the power of CDSA as an easy and versatile tool to control the shape, size, and chemistry of nanoparticles presents many opportunities to tune hydrogel mechanical properties. The ability of the NCs containing polymeric platelet particles to increase the strength of alginate gels, self-heal, and promote adhesion on two freshly cut surfaces of the same gel suggests the attractive possibility of improving calcium-alginate hydrogel performance in vivo and also of adopting them as a method for self-repairing adhesive joints. The biocompatible nature of the materials used provides significant potential for use in a wide range of applications, including tissue engineering and drug delivery.

## Methods

**General**. $^1$H and $^{13}$C nuclear magnetic resonance spectroscopy: $^1$H and $^{13}$C nuclear magnetic resonance (NMR) spectra were recorded at 400 MHz on a Bruker DPX-400 spectrometer in CDCl$_3$ at 298 K, unless otherwise stated. Chemical shifts are reported as $\delta$ in parts per million (ppm) downfield from the internal standard trimethylsilane.

Size-exclusion chromatography: Size-exclusion chromatography (SEC) measurements were performed on a Varian 390-LC-Multi detector suite system fitted with RI and ultraviolet detectors ($\lambda = 309$ nm) equipped with a PLGel 3 μm (50 × 7.5 mm) guard column and two PLGel 5 μm (300 × 7.5 mm) mixed-D columns using DMF with 5 mM NH$_4$BF$_4$ at 50 °C as the eluent at a flow rate of 1.0 mL min$^{-1}$. SEC data were calibrated against PMMA standards and analyzed using Cirrus v3.3 software.

Infrared spectroscopy: Infrared (IR) spectra were recorded on a Perkin Elmer, Spectrum 100 FT-IR Spectrometer at room temperature.

Zeta potential: Zeta potentials were measured using a Malvern Zetasizer Nano. The zeta potentials of particles in suspension at pH 2 were obtained by measuring the electrophoretic movement of the particles under an applied electric field at 25 °C. All determinations were repeated five times.

Dynamic Light Scattering: Hydrodynamic diameters ($D_h$) of particles were determined by dynamic light scattering using a Malvern Zetasizer Nano ZS fitted with a 4 mW He-Ne 633 nm laser module at 25 °C. Measurements were carried out at a detection angle of 173° (back scattering) and the data were analyzed using Malvern DTS 7.03 software. All determinations were repeated five times. $D_h$ was

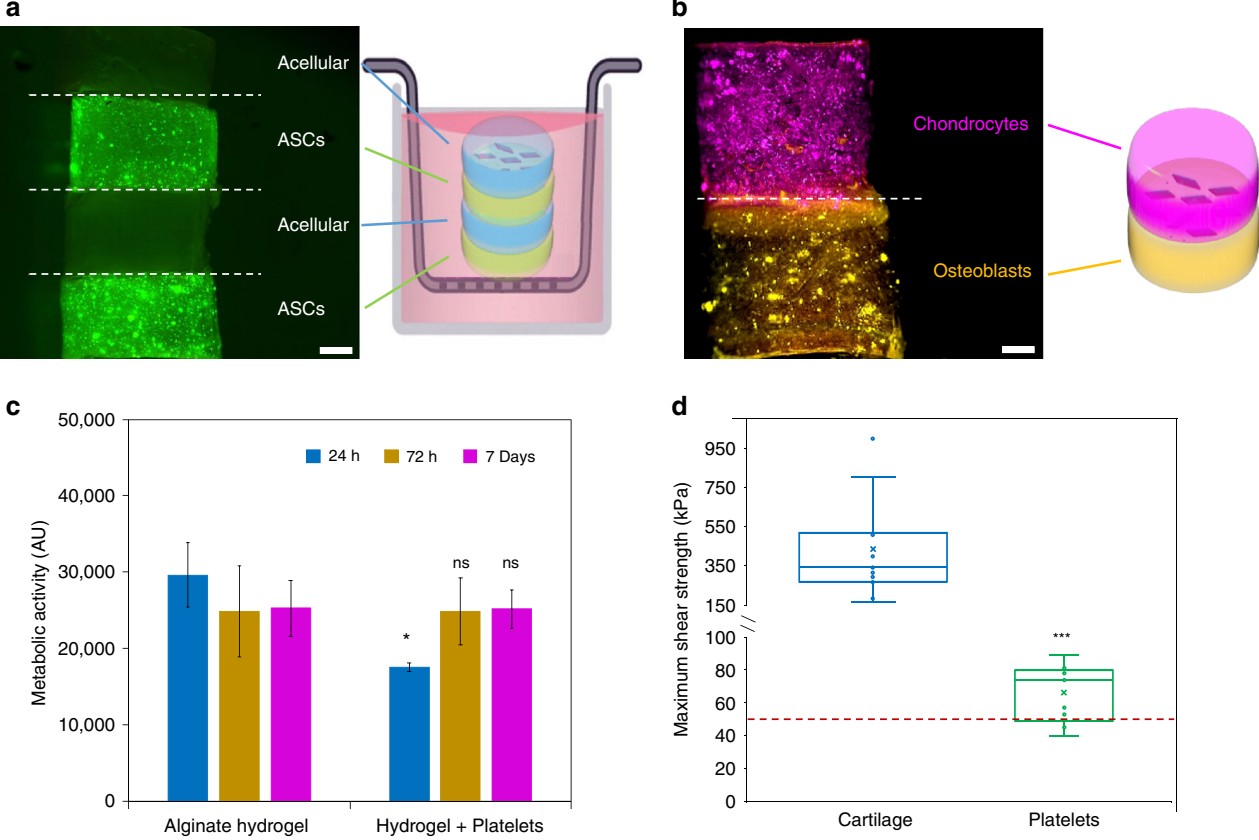

**Fig. 5 Cytocompatibility of calcium-alginate NCs and adhesion of cartilage tissue using PLLA$_{35}$-*b*-PDMAEMA$_{315}$ platelets.** Confocal microscopy images and schematics of calcium-alginate constructs containing **a** both acellular and ASCs encapsulated in alternating layers, glued using PLLA$_{35}$-*b*-PDMAEMA$_{315}$ platelets, with cells stained for live(green)/dead(red) cells. **b** False-color image of labeled chondrocytes (magenta) and osteoblasts (yellow) in separate layers glued together using PLLA$_{35}$-*b*-PDMAEMA$_{315}$ platelets. **c** ASCs proliferation data when encapsulated on alginate hydrogels and PLLA$_{35}$-*b*-PDMAEMA$_{315}$ platelets enriched hydrogels obtained using Alamar Blue metabolic assay. Statistical analysis was performed using a Welch's unpaired *t*-test ($p < 0.05$, $N = 3$). **d** Shear force measurement acquired by lap shear tests on either intact bovine cartilage or cartilage strips adhered with with PLLA$_{35}$-*b*-PDMAEMA$_{315}$ platelets. The red line shows the minimum threshold, from literature, for clinical application in repair of meniscal tears[27]. Statistical analysis was performed using a Welch's unpaired *t*-test ($p < 0.05$, $N = 9$). Error bars represent the standard deviation of the data.

calculated using the Stokes–Einstein equation where particles are assumed to be spherical.

Transmission electron microscopy: Transmission electron microscopy (TEM) was performed using a JEOL 2100 at an acceleration voltage of 200 kV. TEM samples were prepared as follows, unless otherwise stated; 7 µL of a 0.5 mg mL$^{-1}$ solution was deposited onto a formvar grid placed on filter paper and left to dry for at least 45 min. A total of 7 µL of a 1% uranyl acetate solution was dropped onto the grid and left to dry. TEM images were analyzed by ImageJ software, where at least 300 particles were counted for each sample to obtain the length (longest axis), width (shortest axis), and area of the platelet particles and the diameter of spherical particles.

Atomic force microscopy (AFM): Samples for atomic force microscopy (AFM) analysis were prepared by drop casting 7 µL of polymer in ethanol (0.25 mg mL$^{-1}$) onto silicon wafer followed by drying with compressed air. Imaging and analysis were performed on an Asylum Research MFP3D-SA atomic force microscope in alternate contact (tapping) mode.

**Polymer synthesis and NC characterization.** Materials: Chemicals and solvents were purchased from Sigma Aldrich, Acros, Fluka, Fisher Chemical, Alfa Aesar, or VWR. L-Lactide monomer was kindly donated by Corbion-Purac and dried over 3 Å molecular sieves in dichloromethane before recrystallization from toluene and stored in a glove box with inert atmosphere. 1,4-Dioxane was purified by passing through basic alumina before use. (−)Sparteine was dried over calcium hydride and distilled before use. Bis[(trifluoromethyl)phenyl]-3-cyclohexylthiourea was prepared as previously reported[30]. 2,2′-Azobis(2-methylpropionitrile), AIBN, was recrystallized twice from methanol and stored in the dark at 4 °C. Low-molecular weight, low-viscosity sodium alginate was purchased from Sigma Aldrich as a light brown powder (Product code: W201502).

Synthesis of PLLA$_{35}$: In a nitrogen-filled glove box, a solution of 1-[3,5-bis(trifluoromethyl)phenyl]-3-cyclohexylthiourea (120 mg, 0.325 mmol), (−)-sparteine (37.3 µL, 0.162 mmol), and 2-cyano-5-hydroxypentan-2-yl ethyl carbonotrithioate (45 mg, 0.180 mmol) in dry CH$_2$Cl$_2$ (2 mL) were added to a solution of L-lactide (936 mg, 6.495 mmol) in dry CH$_2$Cl$_2$ (8 mL). After stirring for 3 h at room temperature, the solution was removed from the glove box and precipitated three times into ice-cold *n*-hexane. The resultant yellow polymer was filtered and dried under vacuum over P$_2$O$_5$ for two days. $M_{n,NMR} = 5.3$ kDa, DP = 35. $M_{n,SEC} = 9.7$ kDa, Đ$_M = 1.19$. $^1$H NMR (CDCl$_3$): δ (ppm) 5.30 (q, 68H, $^3J_{H-H} = 7.0$ Hz, OC*H*(CH$_3$)CO) 4.31 (q, 1H $^3J_{H-H} = 7.0$ Hz, C*H*(CH$_3$)OH), 4.22 (t, 2H, OC*H$_2$*) 3.37 (t, 2H, $^3J_{H-H} = 6.3$ Hz, SC*H$_2$*) 1.75 (s, 3H, (CN)CC*H$_3$*) 1.20–1.30 (m, 4H, C$_2$*H$_4$*) 1.60 (d, 222H, $^3J_{H-H} = 7.0$ Hz, CH$_3$CHO) 0.88 (t, 3H, $^3J_{H-H} = 6.3$ Hz, CH$_3$CH$_2$).

Synthesis of PLLA$_{35}$-b-PDMAEMA$_{315}$: PLLA$_{35}$ (1 eq.), DMAEMA (500 eq.), and AIBN (0.1 eq.) were dissolved in 1,4-dioxane (1:2 vol% DMAEMA:1,4-dioxane) before transferring to a dried ampoule under nitrogen. After three freeze-pump-thaw cycles, the solution was sealed under nitrogen and heated for 6 h at 70 °C until 60% conversion was reached. The reaction was quenched in liquid nitrogen and purified by precipitation three times into cold diethyl ether. The resultant pale yellow solid was dried in vacuo before use. $M_{n,NMR} = 55.0$ kDa, DP = 315. $M_{n,SEC} = 61.7$ kDa, Đ$_M = 1.27$. $^1$H NMR (CDCl$_3$): δ (ppm) 5.14 (q, 70H, $^3J_{H-H} = 6.9$ Hz, OC*H*(CH$_3$)CO), 4.04 (br s, 756H, OC*H$_2$*CH$_2$CH$_2$N), 2.54 (br s, 748H, OCH$_2$C*H$_2$*N), 2.26 (s, 2336H, N(C*H$_3$*)$_2$), 1.89–1.56 (m, 626H, (CH$_3$)CC*H$_2$*, C*H$_2$*S, CH$_3$CHO), 1.03–0.88 (m, 1100H, C*H$_3$*C(CN)C$_3$*H$_6$*, CH$_3$CH$_2$S, (C*H$_3$*)CCH$_2$).

Preparation of PLLA$_{35}$-b-PDMAEMA$_{315}$ spherical micelles: 18.2 MΩ × cm water (5 mL) was added at to a solution of PLLA$_{35}$-b-PDMAEMA$_{315}$ (10 mg) dissolved in DMF (5 mL) using a peristaltic pump at 0.5 mL h$^{-1}$. The solution was dialyzed against 18.2 MΩ × cm water (MWCO = 3.5 kDa) for 3 days and lyophilized.

Preparation of PLLA$_{35}$-b-PDMAEAm$_{400}$ cylindrical micelles: PLLA$_{35}$-b-PAA$_{400}$ cylindrical micelles were prepared using a combination of ring-opening

polymerization and reversible addition-fragmentation chain transfer polymerization, as reported previously[31,32]. The cylinders (1 eq.) were stirred at 1 mg mL$^{-1}$ at 4 °C at pH 5 for 3 h before addition of sulfo-$N$-hydroxysuccinimide (600 eq.). The dispersion was then stirred for 30 min before addition of ethylcarbodiimide hydrochloride (450 eq.) and stirred a further 3 h. The pH was raised to 7.5 and the dispersion was added to a stirred solution of dimethylethylenediamine (3000 eq.) in water (15 mg mL$^{-1}$) at 4 °C. After stirring overnight at 4 °C, the cylinders were dialyzed against water and lyophilized. $M_{n,NMR}$ = 47.8 kDa, $M_{n,SEC}$ = 45.6 kDa, $Đ_M$ = 1.17. $^1$H NMR (D$_2$O): $\delta$ (ppm) 3.36–3.89 (br s, C(O)N$H$, NHC$H_2$C$H_2$N), 3.21 (br s, NHCH$_2$C$H_2$N), 2.84 (s, N(C$H_3$)$_2$), 1.01–2.33 (m, C$H$C$H_2$, C$H_3$C(CN)C$_3$$H_6$, C$H_3$C$H_2$S).

Preparation of platelets: PLLA$_{35}$-$b$-PDMAEMA$_{315}$ for the preparation of neutral platelets or PLLA$_{3535}$-$b$-PAA$_{400}$ for the preparation of anionic platelets (15 mg) was added to ethanol (3 mL) in a 7 mL vial and sealed. The sample was heated at 90 °C for 2 h before cooling to room temperature. After removing the ethanol under airflow, the platelets were redispersed in water and lyophilized to allow for accurate weighing of the particles. Control over particle size was achieved by introducing different amounts of tetrahydrofurane during the self-assemby in ethanol (0% for small, 6% for medium, and 10% for large platelets).

Quaternization of PLLA$_{35}$-$b$-PDMAEMA$_{315}$ platelets: As previously reported[26], PLLA$_{35}$-$b$-PDMAEMA$_{315}$ platelets (1 eq.) were dispersed in ethanol at 5 mg mL$^{-1}$ and bubbled with nitrogen for ca. 20 min before addition of iodomethane (216 eq.) under nitrogen. The solution was stirred overnight before addition of water and subsequent dialysis against ethanol/water (containing sodium thiosulfate) followed by water only. The platelets were lyophilized and redispersed in water at the desired concentration before use. $^1$H NMR (D$_2$O): $\delta$ (ppm) 4.46 (s, COOC$H_2$CH$_2$N), 3.78 (br s, COOCH$_2$C$H_2$N), 3.22 (br m, CH$_2$CH$_2$N(C$H_3$)$_3$), 1.30–0.70 (m, (C$H_3$)CC$H_2$, C$H_2$S, C$H_3$CHO, C$H_3$C(CN)C$_3$$H_6$ C$H_3$C$H_2$S, (C$H_3$)CC$H_2$).

Zwitterionic modification of PLLA$_{35}$-$b$-PDMAEMA$_{315}$ platelets: As previously reported[26], 2-propanol (1.76 mL) was added dropwise to a stirred solution of PLLA$_{35}$-$b$-PDMAEMA$_{315}$ platelets dispersed in water (1.76 mL) at 25 mg mL$^{-1}$. After addition of 3-bromopropane sulfonic acid sodium salt (1.1 eq. per DMAEMA unit), a solution of sodium hydroxide (0.05 eq. per DMAEMA unit) in 1:1 water/2-propanol was added dropwise before stirring overnight. The solution was dialyzed against water to remove salts. The platelets were then lyophilized and redispersed in water at the desired concentration before use. $^1$H NMR (CDCl$_3$): $\delta$ (ppm) 5.16 (q, $^3$$J_{H-H}$ = 6.9 Hz, OC$H$(CH$_3$)CO), 4.06 (br s, OC$H_2$CH$_2$N), 3.36 (br s, SO$_3$C$H_2$), 2.95 (SO$_3$CH$_2$C$H_2$), 2.58 (br s, C$H_2$N(CH$_3$)$_2$C$H_2$), 2.28 (s, N(C$H_3$)$_2$), 1.75–2.05 (m, (C$H_3$)CC$H_2$, C$H_2$S), 1.75 (d, $^3$$J_{H-H}$ = 7.0 Hz, C$H_3$CHO), 1.05–0.75 (m, C$H_3$C(CN) C$_3$$H_6$, C$H_3$C$H_2$S, (C$H_3$)CC$H_2$).

NC gel formation: Alginate gels were prepared at 1.5 wt% sodium alginate. Before use, sodium alginate (1.0 eq.) was heated in water to 70 °C for 1 h to aid dissolution and subsequently cooled to room temperature. Micelles were dispersed in water for 2 h before stirring with calcium carbonate (0.5 eq.), followed by addition to the sodium alginate solution and vortexing for 1 min. After addition of GDL (1.0 eq.), the gel was again vortexed for 1 min before incubating at room temperature for 2 days.

Gel adhesion measurements: Alginate gels prepared as described above in a 1 mL syringe mold were cut in the middle using a sharp scalpel and immediately 10 µL of particle solution (30 mg mL$^{-1}$) was added to one of the freshly cut end. The other gel piece was then placed on top, with the freshly cut surfaces in contact with the nanoparticle solution. The gel was left healing for 1 h before being immersed in water for swelling tests. Food colorant was mixed with the alginate solution for the preparation of colored gels. Dynamic mechanical analysis (STARe System DMA 1, Mettler Toledo) was used to determine the adhesion strength of gel samples using bars (27 × 7.5 × 1 mm, $l$ × $w$ × $t$) and cylinders (10 × 1 mm, diameter × thickness). Uniaxial tensile testing was performed using the bars, tested in stress strain mode with a force ramp setting of 0 N to 1 N at a rate of 0.01 N min$^{-1}$. Cylinders were used to probe bulk shear adhesive force, tested at 1 Hz for 1 to 1000 µm oscillatory displacement at 1 µm increment. Samples with particles on the surfaces were first treated with 10 µL of particle solution with a concentration of 30 mg mL$^{-1}$ (or in the case of spheres 30 µL from 30 mg mL$^{-1}$, or 10 µL from 105 mg mL$^{-1}$) prior to mounting on the clamps or plates, distributed by micropipette and self-wetting across the surface. Tensile sample ends were then placed together before being gently clamped; cylinders were pressed together after mounting on the plates, with 1.5 mm space remaining between plate surfaces. All testing occurred at room temperature on hydrated samples ($N$ = 7, measurements were taken from distinct samples, each of them was analyzed 3 times).

Polymer density: Polymer density was calculated as follows: ca. 20 mg of PLLA$_{35}$-$b$-PDMAEMA$_{315}$ was weighed accurately into a 1 mL volumetric flask, using a Mettler Toledo XP205 precision balance with five digits (0.00001 g). Methanol was then added to solubilize the sample in aliquots of 100 µL until a total volume of 900 µL was achieved. Subsequently, 10 µL aliquots of MeOH were added until the meniscus reached the 1 mL volume in the volumetric flask. Density was calculated as grams of polymer weighed × polymer volume, where the latter was calculated from 1 mL (volumetric flask volume)—volume added. $\rho$ = 0.804 ± 0.06 g cm$^{-3}$, as average ± standard deviation, where $n$ = 5.

Surface area of spheres and platelets at the adhesion joint: Total surface area (SA$_{total}$) of nanoparticles at the adhesion joint was calculated as outlined below:

$$SA_{total} = SA_{part} \times N_i \quad (1)$$

where: SA$_{part}$ is the surface area of one particle and $N_i$ is the total number of particles.

Surface area of the platelets is calculated as:

$$SA_{part} = 2 \times SA_{face} + 4 \times SA_{side} \quad (2)$$

where: SA$_{face}$ is the surface area of the face of one particle and SA$_{side}$ is the surface area of the side of one particle that are calculated from:

$$SA_{face} = (l_1 \times l_2)/2 \quad (3)$$

$$SA_{side} = h \times \sqrt{\left(\frac{l_1}{2}\right)^2 + \left(\frac{l_2}{2}\right)^2} \quad (4)$$

where: $l_1$ and $l_2$ are point-to-point lengths of the platelets.

Surface area of the spheres is calculated as:

$$SA_{part} = 4\pi r^2 \quad (5)$$

where: $r$ is the radius of a particle.

Surface area of the cylinders is calculated as:

$$SA_{part} = 2\pi rh + 2\pi r^2 \quad (6)$$

where: $r$ is the radius of a particle.

Number of particles, $N_i$, in a sample is calculated as:

$$N_i = \frac{m_t}{m_p} \quad (7)$$

where: $m_t$ is the total mass of polymer added for the adhesion (in this study = 0.3 mg) and $m_p$ is the mass of one particle that can be calculated from:

$$m_p = \rho V \quad (8)$$

in which $\rho$ is the density of the polymer and $V$ is the volume of the particle that can be calculated for platelets by:

$$V = \frac{l_1 l_2}{2} h \quad (9)$$

spheres by:

$$V = \frac{4}{3}\pi r^3 \quad (10)$$

and cylinders by:

$$V = \pi r^2 h \quad (11)$$

Rheological measurements: Oscillatory shear rheology measurements were performed on an Anton Paar Modular Compact Rheometer MCR302. Amplitude sweeps were run at variable strain from 0.01 to 100% at 10 s$^{-1}$ angular frequency, $\omega$. Frequency sweeps were run at variable $\omega$ from 0.1 to 100 rad s$^{-1}$ at 0.5% strain. Results were recorded as an average of three consecutive runs at a constant temperature of 20 °C ($N$ = 3, measurements were taken from distinct samples, each of them were run three times). Stress recovery experiments consisted of three consecutive cycles ($n$ = 3, measurements were taken from the same sample), comprising application of 300% strain for 400 s and 0.1% strain for 700 s, both at an $\omega$ of 10 s$^{-1}$ at 20 °C.

Viability of ASCs encapsulated in alginate gels: Human ASCs were isolated from adipose tissue of patients undergoing hip replacement surgery with full written informed consent. All procedures and experiments were performed in accordance with relevant NHS Health Research Authority National Research Ethics Service and University of Manchester approvals, and in accordance with established procedures[33]. Gels were prepared in a similar fashion as described above, using sterile medium as dispersant instead of water. A 5% solution of sodium alginate was used to reconstitute ASC pellets at 1 × 10$^6$ cells mL$^{-1}$. For a final concentration of 2.5% sodium alginate, CaCO$_3$ (6.26 mg mL$^{-1}$) was prepared along with a GDL solution (22.27 mg mL$^{-1}$) and added to the cell pellet. The cellular/acellular alginate solution (100 µL) was inserted into 24-well cell culture inserts with a 0.4 µm pore PET membrane and left to set at 37 °C for 90 min. Following gelation, stacks were created of ASC-containing layers and acellular layers by using 10 µL of the platelet-based adhesive at a concentration of 30 mg mL$^{-1}$ in deionized water. The constructs were then incubated for the selected time points (24 h, 72 h, and 7 days), with the medium being refreshed every 2 days. Cell proliferation was measured using Alamar Blue proliferation assay (Thermofisher), following the protocol from the supplier. The fluorescence intensity was detected using a Biotek FLx800 Plate Reader ($\lambda_{Ex.}$ = 530 nm, $\lambda_{Em.}$ = 590 nm). Cell data are reported as viability % in comparison with a control sample. Significance was set at a $p$ value ≤ 0.05. Experiments were performed in triplicate ($N$ = 3, measurements were taken from distinct samples). Cell viability was also measured using live/dead kit with calcein staining (live cells, green, $\lambda_{Ex.}$ = 495 nm, $\lambda_{Em.}$ = 515 nm) and ethidium homodimer (dead cells, red, $\lambda_{Ex.}$ = 528 nm, $\lambda_{Em.}$ = 617 nm), following the protocol provided by the supplier. All images were captured on a Leica M205 FA Upright Stereofluorescence microscope, using a 488 nm laser for the green channel and a 514 nm laser for the red channel.

Chondrocytes and osteoblasts encapsulation in alginate gels: Human chondrocyte (T/C-28a2) and osteoblast (SaOS2) cell lines were used to model the

osteochondral junction. Cells were fluorescently labeled using PKH26 red and PKH67 green fluorescent cell linker mini kit for general cell membrane labeling (Sigma Aldrich). Cell staining of osteoblasts (PKH26) and chondrocytes (PKH67) was performed as per manufacturer's instructions. Following cell labeling, cell pellets containing $1 \times 10^6$ cells were reconstituted in 5% sodium alginate. For a final concentration of 2.5% sodium alginate, $CaCO_3$ (6.26 mg mL$^{-1}$) was prepared along with a GDL solution (22.27 mg mL$^{-1}$) and added to the cell pellet. The cellular alginate solution (200 μL) was inserted into 24-well cell culture inserts with a 0.4 μm pore PET membrane and left to set at 37 °C for 90 min. Following gelation, stacks of osteoblast- and chondrocyte-containing layers were constructed using 10 μL of the platelet-based adhesive at a concentration of 30 mg mL$^{-1}$ in deionized water. After 30 min, stacks were submerged in growth medium. Images were captured on a Leica M205 FA Upright Stereofluorescence microscope, using FITC ($\lambda_{Ex.} = 490$ nm, $\lambda_{Em.} = 525$ nm) and TRITC ($\lambda_{Ex.} = 556$ nm, and $\lambda_{Em.} = 563$ nm) channels, and images false-colored magenta (chondrocytes) and yellow (osteoblasts).

Bovine cartilage adhesion: Two overlapping strips of bovine articular cartilage (30 mm length × 5 mm width × 2 mm depth) were adhered together at the synovial surface using 50 μL PLL-DP (120 mg mL$^{-1}$) and allowed to dry at room temp for 10 min. A 10 mm length of adhered cartilage, at both ends of the strip, was used to clamp the strip into the electroforce machine. The area of adhesion was 75 mm$^2$. The adhesive strength of the PLL-DP was evaluated by measuring the maximum shear force (N) generated by a 200 N load cell (TA electroforce 5500) at an extension rate of 5 mm min$^{-1}$. The strength at failure was recorded for each sample ($N = 9$, measurements were taken from distinct samples).

Statistical analysis: Statistical analysis has been performed on the adhesion (both of hydrogel and tissue surfaces) and rheology measurements and on the cell viability data using a two-way ANOVA test or a Welch's unpaired $t$-test, with $p < 0.05$, analyzed with GraphPrism 8 software.

**Reporting summary**. Further information on research design is available in the Nature Research Reporting Summary linked to this article.

## Data availability

The authors declare that the data supporting the findings of this study are available within the paper and its Supplementary Information file. Raw data underlying the figures presented are available from the corresponding authors upon reasonable request.

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

## Acknowledgements

The University of Warwick is thanked for the award of a DTP studentship (M.I.). This project has received funding from the European Research Council (ERC) under the European Union's Horizon 2020 research and innovation programme under grant agreement numbers 681559 and 615142 to support A.P.D., R.K.O., and M.C.A. This work was supported by the Whitaker Foundation to support A.C.W. The Medical Research Council are thanked for funding C.I.P. via a Confidence-in-Concept 2016 award to The University of Manchester (MC_PC_16053). This work was also supported by the Henry Royce Institute for Advanced Materials, funded through EPSRC grants EP/R00661X/1 and EP/P025021/1. All three reviewers are thanked for their time and contribution to the final version of this paper.

## Author contributions

M.I. and M.C.A. contributed equally to this work. M.C.A., A.P.D., and R.K.O. designed the study, supervised the research, and wrote the paper. M.I. and Z.L. prepared the NC gels and carried out rheological measurements. A.C.W. designed and carried out the gel adhesion measurements. A.L.A.B. carried out the cell encapsulation experiments, and C.I.P. carried out the tissue adhesion experiments with supervision from S.M.R. and J.A.H. All authors approved the final version of the manuscript.

## Competing interests

The authors declare no competing interests.
