## [Peer Review File · Nature Communications]

Reviewers' comments:

Reviewer #1 (Remarks to the Author):

Summary:

This manuscript describes the ability to enhance hydrogel adhesion properties and alter key mechanical properties using nanoparticle additives based on soft matter. The relationships between nanoparticle morphology (spheres, cylinders, platelets), as well as surface functionalization (i.e. nanoparticle charge), and the resulting nanocomposite hydrogel properties were studied. This methodology was also translated into bovine cartilage tissue, demonstrating the potential translation of this system into bio-relevant applications. Overall, the work is of exceptional quality, significance, and current relevance and interest. Publication in Nature Communications is recommended after addressing the following points.

Specific comments:

1. Page 1 / line 6: poly(L-lactide) not poly(l-lactide)
2. Page 2/ Figure 1 and associated text. Dispersity information is given for the platelets but not for the spheres and cylinders. From Figure 1 the spheres look monodisperse but the cylinders look polydisperse. Please add information on dispersity, especially for the cylinders as this is important.
3. Page 4 / line 14: expand on argument why the cylinders elongated morphology provides less interaction with the gels compared with spheres, considering earlier argument of surface area of the nanoparticles being a key consideration. 'Cylinders interact less with gels leading to a lower strain at break' because surface area is smaller than platelets? But why do spheres have a higher strain at break if their surface area is smaller?
4. Page 5 / line 5: give a justification for why the strain at yield is nanoparticle loading-dependent for platelet morphology, but not for spherical or cylindrical micelles
5. Page 5 Figure 3E - error bars appear to be missing in the region where the medium and large platelets overlap (NP content 0.04-0.06 wt %). They are important to explain the overlap.
6. Page S1 / Synthesis of PLLA35: Mn by 1H NMR is shown to be 5.4 kDa, DP = 36. Mn by GPC is shown to be even higher. Throughout text PLLA is referred to have DP = 35. Should keep this consistent (perhaps with NMR data)
7. Page S7 / Figure S1: integration of peak 'g' corresponds to DP = 34
8. Page S6-S7 (WAXD and SAED): No WAXD or SAED data was shown in manuscript or SI - delete the description of these two instruments
9. (IMPORTANT) The key synthetic experimental method that allows dimensional control of the platelets is not very clear in the supporting information, Under "Typical preparation of platelets" on page S2 the text states that precursor neutral and zwitterionic platelet samples are heated at 90 °C for 2 h in various solvent compositions to achieve size control. However, where these original precursor samples come from is not made clear. Are these initial precursors polydisperse and how are they made ? Please add the necessary i) synthetic information and also ii) characterization of their dimensions and dispersity.

Reviewer #2 (Remarks to the Author):

The manuscript by Dove and O'Reilly entitled "Exploiting the Role of Nanoparticle Shape in Enhancing Hydrogel Adhesive and Mechanical Properties" describes the use and affect of nanoparticle shape, when added to an alginate gel, on gel adhesion and strength. Sphere, rod, and platelets shapes were evaluated. The gel nanocomposites containing platelets showed enhanced resistance to breaking under strain as well as increased adhesion strength compared to their spherical and cylindrical counterparts. Using an optimized formulation they showed the ability to seal cut cartilage tissue - suggesting its potential use in the clinic.

The manuscript is well written and data support the conclusions.

I enjoyed reading this manuscript and the results will be of interest to the broader community. By varying morphology, size and charge the authors are able to provide information on what nanoparticle works best and why. The surface area experiments between spheres and platelets were particularly nice.

I recommend publication after the following changes are made and minor comments addressed.

1. Increasing the amount of platelets added to the gel over a range of 0.04 to 0.12 wt% improved performance (Figure 3D). What happens if you add substantially more material such as 1 or 10 wt%? Is there an upper bound to the enhancement.
2. The text or SI does not provide information on the alginate used in the experiments. Where was the alginate purchased? What is the molecular weight and/or viscosity? What percentage of G vs M blocks are present in the material?
3. Please provide statistical analysis of the data, include p values and n or N values in the text and figures. What data are significant and what data are not?
4. For the adhesion energy experiments (Figure 2D) – please provide the control gel without particles in the graph. Additionally, please provide further discussion on the magnitude of these adhesion energy values compared to familiar materials such a commercial cyanoacrylate or PEG or fibrin based adhesive. This will help the reader understand the potential uses of these materials.
5. For the sentence “A construct with alternate layers of encapsulated ASCs gel and acellular gel was built using the PDMAEMA-functional platelets as an adhesive between layers..” is the PDMAEMA-functional platelets the same as the PLLA35-b-PDMAEMA315 platelets - please be consistent in the terminology throughout the manuscript.
6. For the cell viability experiments with the ASCs (adipose-derived stem cells), please provide a more quantitative assessment of cell viability besides a confocal microscopy image. Are you able to dissolve the hydrogel and perform a flow experiment or standard MTS assay on the cells after 7 days?
7. For the shear force measurements with cartilage strips (Figure 5D), the control group of alginate without platelets is missing in the figure. Please provide these data.
8. Please check the reference and sentence “Nevertheless, this result suggests that an optimized nanoparticle system of this kind could be used to glue cartilage tissues, replacing the toxic cyanoacrylates currently employed.²⁷” This sentence suggests that cyanoacrylates are used clinically for this procedure. I believe they are not FDA approved for internal use. Please clarify.
9. The ASSOCIATED CONTENT states “The Supporting Information is available free of charge on the ACS Publications website.” Please edit.

Reviewer #3 (Remarks to the Author):

The manuscript by Arno et al. describes the adhesive properties of nanoparticles with different surface chemistries and geometries to link hydrogel blocks and biological tissues together. The

concept of varying chemical features and the shape of nanoparticles to modulate their adhesive properties is not novel (Liu, H., Peng, Y., Yang, C., Wang, M., Part. Part. Syst. Charact. 2017, 34, 1700286). The authors report that platelet-shaped quaternized nanoparticles show improved adhesive properties towards alginate hydrogels, when compared to spherical and cylindrical micelles with comparable zeta potential values. This effect is ascribed to a combined effect of electrostatic interactions between alginate molecules and the nanoparticles, as well as to the ability of these nanoparticles to spread evenly and adjust to contours in the interface established between two biomaterial blocks. Although this is an interesting postulation, the study lacks characterization to establish an effective proof for this theory. Moreover, the height of platelets is not provided, and it is unclear whether this parameter changes for platelets with different planar areas. Overall, although the results presented in this manuscript are somewhat interesting, the concept of varying nanoparticles' size and chemical features to modulate adhesive properties is not new, and extensive characterization would be needed to enlighten underlying mechanisms.

Reviewers' comments:

Reviewer #1 (Remarks to the Author):

Summary:

This manuscript describes the ability to enhance hydrogel adhesion properties and alter key mechanical properties using nanoparticle additives based on soft matter. The relationships between nanoparticle morphology (spheres, cylinders, platelets), as well as surface functionalization (i.e. nanoparticle charge), and the resulting nanocomposite hydrogel properties were studied. This methodology was also translated into bovine cartilage tissue, demonstrating the potential translation of this system into bio-relevant applications. Overall, the work is of exceptional quality, significance, and current relevance and interest. Publication in Nature Communications is recommended after addressing the following points.

Specific comments:

1. Page 1 / line 6: poly(L-lactide) not poly(l-lactide)

This formatting error has now been corrected.

2. Page 2/ Figure 1 and associated text. Dispersity information is given for the platelets but not for the spheres and cylinders. From Figure 1 the spheres look monodisperse but the cylinders look polydisperse. Please add information on dispersity, especially for the cylinders as this is important.

We apologise for the confusion. All dispersity data are now provided in Figure 1 caption, Table S2 and S3 of the supplementary information.

3. Page 4 / line 14: expand on argument why the cylinders elongated morphology provides less interaction with the gels compared with spheres, considering earlier argument of surface area of the nanoparticles being a key consideration. 'Cylinders interact less with gels leading to a lower strain at break' because surface area is smaller than platelets? But why do spheres have a higher strain at break if their surface area is smaller?

We thank the reviewer for bringing out this point, as it offers the opportunity to further clarify this concept, both in this response and in the manuscript. The surface area for of the cylinders, per particle, ($9.6 \times 10^{-11} \pm 1.6 \times 10^{-11} \text{ cm}^2$) is actually smaller compared to that of spheres ($7.1 \times 10^{-10} \pm 3.4 \times 10^{-10} \text{ cm}^2$) and platelets (small: $9.3 \times 10^{-10} \pm 8.7 \times 10^{-10} \text{ cm}^2$, medium: $4.9 \times 10^{-9} \pm 5.0 \times 10^{-8} \text{ cm}^2$, and large: $1.7 \times 10^{-8} \pm 6.2 \times 10^{-7} \text{ cm}^2$), therefore offering less surface area per particle to interact with the alginate. However, the surface area of small platelets and spheres is similar (9.3×10^{-10} vs. $7.1 \times 10^{-10} \text{ cm}^2$) but the strain at break of the platelets is much higher (70%) compared to the spheres (25%). This further validates our hypothesis that not only surface area, but particle shape is a key element to consider in enhancing the mechanical properties of alginate gels. We postulate that, as a consequence of their 2 dimensional morphology, platelets offer a greater available number of possible anchor points compared to the 0D morphology, and have therefore a greater effect on the hydrogel's mechanical properties.

These data have now been added in Table S2 and the text has been modified with the following: "This is not surprising, as the elongated morphology of cylindrical micelles is expected to interact less with the alginate gels, as a consequence of the lower surface area per particle (9.6×10^{-11} vs. $7.1 \times 10^{-10} \text{ cm}^2$), leading to a lower strain at break". And later: "We postulate that, as a consequence of their 2D morphology, platelets offer a greater surface area

that is more easily available to interact with the hydrogel's polymeric chains, therefore further enhancing the mechanical properties of alginate gels compared to both spherical and cylindrical morphologies".

4. Page 5 / line 5: give a justification for why the strain at yield is nanoparticle loading-dependent for platelet morphology, but not for spherical or cylindrical micelles

The observed relationship between particle loading and hydrogel strain at flow point in the case of the platelet morphology can be ascribed to the higher interaction of the 2D platelets with the alginate hydrogel, hence the more platelets are incorporated in the gel the higher the strain at flow point. We believe that the effect of spherical and especially cylindrical micelles is so minimal that even increasing the amount of nanoparticles in the system does not translate in an increase in the strain at flow point. The text has been modified to include this explanation.

5. Page 5 Figure 3E - error bars appear to be missing in the region where the medium and large platelets overlap (NP content 0.04-0.06 wt %). They are important to explain the overlap.

The error bars are present in the graph for every point examined, however the authors recognise these are too small to be clearly visible as indicated by this reviewer. The size of the markers was reduced to make the error bars more visible and the data have been added in Table S9 in the supplementary information.

6. Page S1 / Synthesis of PLLA₃₅: M_n by ¹H NMR is shown to be 5.4 kDa, DP = 36. M_n by GPC is shown to be even higher. Throughout text PLLA is referred to have DP = 35. Should keep this consistent (perhaps with NMR data)

The M_n calculated by NMR has been corrected in this section to 5.3 kDa (DP = 35). The M_n value obtained by SEC does not represent of the true molecular weight of the PLLA sample; as specified in the characterisation methods section in the supplementary information, our SEC was calibrated using poly(methylmethacrylate) which retention time significantly differ from PLLA, hence the molecular weight obtained using this method is not as accurate.

7. Page S7 / Figure S1: integration of peak 'g' corresponds to DP = 34

We apologise for the confusion and thank the reviewer for noticing this mistake. The error arises from an uncorrected assignment of the PLLA NMR spectrum, which has now been rectified. The DP of PLLA corresponds to 35.

8. Page S6-S7 (WAXD and SAED): No WAXD or SAED data was shown in manuscript or SI – delete the description of these two instruments

We thank the reviewer for noticing this oversight. These sections have now been deleted.

9. (IMPORTANT) The key synthetic experimental method that allows dimensional control of the platelets is not very clear in the supporting information, Under "Typical preparation of platelets" on page S2 the text states that precursor neutral and zwitterionic platelet samples are heated at 90 °C for 2 h in various solvent compositions to achieve size control. However, where these original precursor samples come from is not made clear. Are these initial precursors polydisperse and how are they made? Please add the necessary i) synthetic information and also ii) characterization of their dimensions and dispersity.

We apologise for the confusion. The precursor the reviewer refers to in this case is the polymer PLLA₃₅-b-PDMAEMA₃₁₅ or PLLA₃₅-b-PAA₄₀₀ prepared as discussed in the methods section (Synthesis of PLLA₃₅-b-PDMAEMA₃₁₅) and in our previous reports (Chem.

Sci. 2011, 2 (5), 955-960; Soft Matter 2012, 8 (28), 7408-7414), respectively. The text has been modified as follows: “PLLA₃₅-b-PDMAEMA₃₁₅ for the preparation of neutral platelets or PLLA₃₅-b-PAA₄₀₀ for the preparation of anionic platelets (15 mg) was added to ethanol (3 mL) in a 7 mL vial and sealed” to make this clearer.

Reviewer #2 (Remarks to the Author):

The manuscript by Dove and O’Reilly entitled “Exploiting the Role of Nanoparticle Shape in Enhancing Hydrogel Adhesive and Mechanical Properties” describes the use and affect of nanoparticle shape, when added to an alginate gel, on gel adhesion and strength. Sphere, rod, and platelets shapes were evaluated. The gel nanocomposites containing platelets showed enhanced resistance to breaking under strain as well as increased adhesion strength compared to their spherical and cylindrical counterparts. Using an optimized formulation they showed the ability to seal cut cartilage tissue – suggesting its potential use in the clinic.

The manuscript is well written and data support the conclusions.

I enjoyed reading this manuscript and the results will be of interest to the broader community. By varying morphology, size and charge the authors are able to provide information on what nanoparticle works best and why. The surface area experiments between spheres and platelets were particularly nice.

I recommend publication after the following changes are made and minor comments addressed.

1. Increasing the amount of platelets added to the gel over a range of 0.04 to 0.12 wt% improved performance (Figure 3D). What happens if you add substantially more material such as 1 or 10 wt%? Is there an upper bound to the enhancement.

When the content of platelets in the gel was raised above 0.12 wt% a decrease in strain at flow point was observed, likely as an increased competition of ionic interactions between micelle-alginate vs. calcium alginate, resulting in a disruption of the hydrogel matrix. This point has been discussed in the manuscript in the following paragraph: “As expected, a maximum limit was reached (0.12 wt%) where the gels showed decreased homogeneity, presumably as a consequence of the increased competition between micelle-alginate and calcium-alginate ionic interactions becoming important and hence disrupting gel formation”.

2. The text or SI does not provide information on the alginate used in the experiments. Where was the alginate purchased? What is the molecular weight and/or viscosity? What percentage of G vs M blocks are present in the material?

We apologise for the missing information. The following details were added in the “materials” section in the SI: “Low Molecular weight, low viscosity sodium alginate was purchased from Sigma Aldrich as a light brown powder (Product code: W201502)”. Unfortunately, the supplier did not provide any information on the G/M ratio in the material.

3. Please provide statistical analysis of the data, include p values and n or N values in the text and figures. What data are significant and what data are not?

Statistical analysis has been performed on the adhesion (both of hydrogel and tissue surfaces) and rheology measurements and on the cell viability data using a two-way ANOVA test or a Welch’s unpaired t test, with $p < 0.05$; n and N values have been specified for all samples in

the figure captions. Statistical analysis of the bulk shear adhesive energy of alginate hydrogels adhered with PLLA₃₅-based nanoparticles shows that small cationic platelets are significantly superior in enhancing hydrogel adhesion compared to spherical and cylindrical micelles (Table S7), medium and large platelets (Table S5), and anionic, zwitterionic, and neutral platelets (Table S6). Similarly, statistical analysis of the strain at flow point of alginate hydrogels enriched with different concentrations of PLLA₃₅-based nanoparticles shows that small cationic platelets are significantly superior in enhancing the hydrogels' mechanical properties compared to spherical and cylindrical micelles (Table S9) or medium and large platelets (Table S10). Finally, no statistical difference was observed between the viability of ASCs after 7 days encapsulation in alginate hydrogels (control) and hydrogels enriched with PLLA₃₅-b-PDMAEMA₃₁₅ platelets (Figure 5C).

4. For the adhesion energy experiments (Figure 2D) – please provide the control gel without particles in the graph. Additionally, please provide further discussion on the magnitude of these adhesion energy values compared to familiar materials such a commercial cyanoacrylate or PEG or fibrin based adhesive. This will help the reader understand the potential uses of these materials.

The control gel without particles has been added in graph 2D. As suggested by this reviewer, the adhesive strength of the commercial cyanoacrylate Dermabond, a PG-based adhesive, and the commercial fibrin glue Tisseel have been added to the discussion and the text was modified as following: “Nevertheless, the adhesive energy provided by our platelets is higher than the adhesive energy reported for PEG-DOPA and commercial fibrin adhesive Tisseel (35 ± 12.5 and 2.58 ± 1.76 kPa respectively), although lower than the commercial cyanoacrylate Dermabond (181 ± 33.4 kPa)”.

5. For the sentence “A construct with alternate layers of encapsulated ASCs gel and acellular gel was built using the PDMAEMA-functional platelets as an adhesive between layers.” is the PDMAEMA-functional platelets the same as the PLLA₃₅-b-PDMAEMA₃₁₅ platelets - please be consistent in the terminology throughout the manuscript.

These has been corrected. Moreover, the whole manuscript has been checked to ensure consistency in the terminology of the nanoparticle systems used in this work.

6. For the cell viability experiments with the ASCs (adipose-derived stem cells), please provide a more quantitative assessment of cell viability besides a confocal microscopy image. Are you able to dissolve the hydrogel and perform a flow experiment or standard MTS assay on the cells after 7 days?

These data are already present in the manuscript, as Figure 5C. The graph and caption have been modified to make it clearer to the reader that the cell viability data come from an Alamar Blue proliferation assay performed on ASCs encapsulated in the alginate gel matrix with or without PLLA₃₅-b-PDMAEMA₃₁₅ platelets for 24 h, 72 h, and 7 days. A detailed description of the experimental methods can be found in the supplementary information, under the section “Viability of ASCs encapsulated in alginate gels”. We apologise for the confusion.

7. For the shear force measurements with cartilage strips (Figure 5D), the control group of alginate without platelets is missing in the figure. Please provide these data.

The data in Figure 5D refer to the bovine cartilage strips cut in two pieces which are then adhered back together using a solution of PLLA₃₅-b-PDMAEMA₃₁₅ platelets. No alginate was involved in this adhesion experiment. All adhesive data for alginate hydrogels are present in Figure 2. A control for this experiment (i.e. cartilage strips adhered back together

with pure water used as adhesive) could not be included as the cartilage strips do not adhere at all in this case. Since the loading rig requires to position the cartilage vertically between two clamps, no resistance could be measured when the tissue is pulled apart, obtaining a meaningless value for this experiment.

8. Please check the reference and sentence “Nevertheless, this result suggests that an optimized nanoparticle system of this kind could be used to glue cartilage tissues, replacing the toxic cyanoacrylates currently employed.”²⁷ “This sentence suggests that cyanoacrylates are used clinically for this procedure. I believe they are not FDA approved for internal use. Please clarify.

We agree with the reviewer that this sentence is misleading. The reference to cyanoacrylates was removed and the text modified as following: “Nevertheless, the adhesive energy provided by our platelets is higher than the adhesive energy reported for PEG-DOPA and commercial fibrin adhesive Tisseel (35 ± 12.5 and 2.58 ± 1.76 kPa respectively), although lower than the commercial cyanoacrylate Dermabond (181 ± 33.4 kPa)”.

9. The ASSOCIATED CONTENT states “The Supporting Information is available free of charge on the ACS Publications website.” Please edit.

This has been corrected.

Reviewer #3 (Remarks to the Author):

The manuscript by Arno et al. describes the adhesive properties of nanoparticles with different surface chemistries and geometries to link hydrogel blocks and biological tissues together. The concept of varying chemical features and the shape of nanoparticles to modulate their adhesive properties is not novel (Liu, H., Peng, Y., Yang, C., Wang, M., Part. Part. Syst. Charact. 2017, 34, 1700286).

In the paper cited by this reviewer, 0D and 1D silica nanoparticles were tested as adhesives between two pieces of liver tissue and their adhesive strength was compared with commercial silica LUDOX. The adhesive strength of both nanoparticle morphologies was negligible compared to LUDOX and no difference was observed between the spherical and cylindrical morphologies; functionalisation with NaOH significantly improved the adhesion properties of both 0D and 1D nanoparticles. However, the use of NaOH destroyed the assemblies, hence the effect of nanoparticle morphology could not be evaluated using this system.

The novelty of our work is focussed on the use of a polymeric system to achieve 0D, 1D, and 2D morphologies using the same polymer chemistry each time. This allows the direct comparison between the three morphologies and hence the evaluation of their effect not only on the adhesion of alginate hydrogels blocks and bovine cartilage tissue, but also on enhancing the mechanical strength of alginate hydrogels thus improving their performance and translational relevance. Moreover, the maximum adhesive strength that bovine cartilage adhered with PLLA₃₅-b-PDMAEMA₃₁₅ platelets can sustain (~ 5000 mN) is more than 4 times higher than the silica nanoparticles described by Wang et al (~ 1100 mN). Thus, we believe that our report brings sufficient novelty and greatly advances the field compared to previous literature.

The authors report that platelet-shaped quaternized nanoparticles show improved adhesive properties towards alginate hydrogels, when compared to spherical and cylindrical micelles with comparable zeta potential values. This effect is ascribed to a combined effect of electrostatic interactions between alginate molecules and the nanoparticles, as well as to the

ability of these nanoparticles to spread evenly and adjust to contours in the interface established between two biomaterial blocks. Although this is an interesting postulation, the study lacks characterization to establish an effective proof for this theory.

The enhancement of the hydrogel's adhesive properties provided by the platelets can be ascribed to different factors, including larger single surfaces per particle as well as a greater number of anchor points compared to cylindrical and spherical morphologies. In response to this and comments from reviewer 1, we have further emphasised this concept in the manuscript.

While the concept of studying the effect of nanoparticle shape on the adhesive properties of hydrogels is novel, spherical silica nanoparticles have been investigated for this purpose. A report from Rose et al. (Nature, 2014, 505, 382-385) hypothesises that the mechanism underneath the adhesion properties of nanoparticles relies on their "ability to adsorb onto the polymer gels and act as connectors between polymer chains, anchoring to the gel network by numerous attachments". Despite the fact that this theory still remains to be experimentally confirmed, perhaps as a consequence of the lack of a methodology that can be used to assess this hypothesis, we expect the same mechanism to be applicable to our system. With this in mind, clearly platelets offer a greater surface available to interact with the hydrogel's polymeric chains as a consequence of their 2D morphology, therefore further enhancing the mechanical properties of alginate gels compared to both spherical and cylindrical morphologies.

Moreover, the height of platelets is not provided, and it is unclear whether this parameter changes for platelets with different planar areas.

The height of the platelets has now been reported in Table S2, as it can be observed from these values this parameter does not significantly change for platelets with different sizes.

Overall, although the results presented in this manuscript are somewhat interesting, the concept of varying nanoparticles' size and chemical features to modulate adhesive properties is not new, and extensive characterization would be needed to enlighten underlying mechanisms.

Concerning this final comment, we believe that these concerns have been addressed in the answers provided above.

REVIEWERS' COMMENTS:

Reviewer #1 (Remarks to the Author):

This is an excellent manuscript which has been further improved by the revisions. The paper is now ready to publish in my view as all of my comments have now been addressed.

Reviewer #2 (Remarks to the Author):

The authors have addressed my previous comments and the manuscript is improved. I recommend publication of the manuscript

With regards to reviewer #3 comments. The authors have addressed the comments and the manuscript is improved. I recommend publication of the manuscript after the addition of the reference by Liu, H., Peng, Y., Yang, C., Wang, M., Part. Part. Syst. Charact. 2017, 34, 1700286.- maybe include it after ref 22.

REVIEWERS' COMMENTS:

Reviewer #1 (Remarks to the Author):

This is an excellent manuscript which has been further improved by the revisions. The paper is now ready to publish in my view as all of my comments have now been addressed.

We thank this reviewer for the valuable comments.

Reviewer #2 (Remarks to the Author):

The authors have addressed my previous comments and the manuscript is improved. I recommend publication of the manuscript

We thank this reviewer for the valuable comments.

With regards to reviewer #3 comments. The authors have addressed the comments and the manuscript is improved. I recommend publication of the manuscript after the addition of the reference by Liu, H., Peng, Y., Yang, C., Wang, M., Part. Part. Syst. Charact. 2017, 34, 1700286.- maybe include it after ref 22.

This reference has now been added as suggested.